# The Phosphate/Urea Nitrogen Ratio in Urine—A Method to Assess the Relative Intake of Inorganic Phosphate

**DOI:** 10.3390/nu17213323

**Published:** 2025-10-22

**Authors:** Carlos Novillo, Raquel M. García-Saez, Laura Sánchez-Molina, Cristian Rodelo-Haad, Andrés Carmona, Gonzalo Pinaglia-Tobaruela, Cristina Membrives-González, Daniel Jurado, Rafael Santamaría, Juan R. Muñoz-Castañeda, Alejandro Martín-Malo, Mariano Rodríguez, Sagrario Soriano, Victoria Pendón-RuizdeMier

**Affiliations:** 1Unidad de Gestión Clínica Nefrología, Reina Sofia University Hospital, 14004 Cordoba, Spain; carlosnov13@hotmail.com; 2Instituto Maimónides de Investigación Biomédica de Córdoba (IMIBIC), Unidad de Gestión Clínica Nefrología, Reina Sofia University Hospital, Department of Medicine, University of Cordoba, 14004 Cordoba, Spain; raquel.garcia@imibic.org (R.M.G.-S.); crisroha@yahoo.com (C.R.-H.); andres.carmona@imibic.org (A.C.); gonzalo.pinaglia@imibic.org (G.P.-T.); daniel.jurado@imibic.org (D.J.); rsantamariao@gmail.com (R.S.); amartinma@senefro.org (A.M.-M.); marianorodriguezportillo@gmail.com (M.R.); marias.soriano.sspa@juntadeandalucia.es (S.S.); mvictoriaprm@gmail.com (V.P.-R.); 3Department of Medicine, University of Cordoba, 14004 Cordoba, Spain; lauraasm71998@gmail.com; 4Redes de Investigación Cooperativa Orientadas a Resultados en Salud (RICORS), Instituto de Salud Carlos III, RD24/0004/0004, 28029 Madrid, Spain

**Keywords:** inorganic phosphate, urinary phosphate/urea, normal renal function, diet, chronic kidney disease

## Abstract

Background/Objectives: It would be desirable to reduce the intake of inorganic phosphate (P), which is easily absorbed and is associated with cardiovascular disease. The phosphate-to-urea nitrogen ratio (P/UUN) in urine should reflect the proportion of inorganic P ingested relative to protein intake. In this manuscript, we will refer to this parameter as P/UUN, which is conceptually equivalent to the phosphate-to-urea ratio (P/U). These studies aim to evaluate whether an increased intake of inorganic P translates into an increase in the P/UUN. Methods: A total of 18 healthy volunteers (Study-1) and 18 chronic kidney disease patients (Study-2) were included. At baseline, all participants completed a 3-day dietary survey, and on the third day collected a 24 h urine sample. In Study-2, blood samples were also obtained. Participants were then stratified into three groups (6 per group) for a 3-day dietary intervention: control group: maintained their usual diet; soda group: consumed soda zero, a source of added inorganic P; and processed cheese group: consumed fresh processed cheese, which includes both organic and inorganic P additives. At last visit, all participants again completed a 3-day dietary survey and collected a 24 h urine sample (and blood samples in Study-2). Dietary P intake was estimated using two tools: the diet calibrator and the Spanish Food Composition Database (BEDCA). Results: After the intervention, neither BEDCA nor the diet calibrator was able to provide an accurate measurement of inorganic P ingested. However, only in the soda group, P/UUN increased in both studies (*p* = 0.046 and 0.047). In Study-2, the levels of FGF23 and klotho remained unchanged (*p* = 0.9 and *p* = 0.7, respectively). Conclusions: These findings suggest that urinary P/UUN ratio may be a useful biomarker to monitor changes in inorganic P intake and could help to individualize dietary recommendations to reduce inorganic P exposure without restricting protein intake.

## 1. Introduction

“The Global Burden of Disease” estimates that chronic kidney disease (CKD) was the fastest growing cause of death, after Alzheimer’s disease, and it was also the cause of “years lived with disability” that increased mainly during the last decade [1]. CKD is progressive and affects >10% of the general population worldwide; it is more prevalent in older individuals, women and in patients with diabetes and hypertension [2]. The adverse impact of CKD should enhance efforts for better prevention and treatment. These patients progressively lose renal function and it would be appropriate to apply as many strategies as possible to reduce the rate of decline in renal function. Previous work by our group has shown that in experimental models of animals with kidney failure and in CKD patients, the high values of the ratio phosphate/creatinine (P/Cr) in urine are associated with the worsening of kidney function [3]. In the earlier stages of CKD, the serum P concentration remains within the normal range because a reduction in kidney filtration of P is compensated by a reduction in tubular reabsorption of P which is mediated by the action of fibroblast growth factor-23 (FGF23) and parathyroid hormone (PTH); both hormones are increased even in moderate renal failure [3,4,5]. There is an association between elevated serum P concentrations and increased mortality in patients with advanced CKD and this is also observed, although less frequently, in the general population [6,7,8,9].

An appropriate strategy would be the reduction in P intake. Numerous elements in our diet contain P, primarily proteins and food additives. The P contained in animal proteins is hydrolyzed and absorbed in the gastrointestinal tract, while the P contained in vegetable proteins is not easily absorbed because it is bound to phytate and humans lack phytase, the enzyme responsible for separating P from phytate [10]. Therefore, the intestinal absorption of P contained in animal protein ranges between 50 and 70%, whereas the absorption of P from vegetable protein is only 30–50%. By contrast the intestinal absorption of inorganic P is close to 100% [10,11,12].

A reduction in dietary proteins will certainly decrease the intake of P but it may interfere with an appropriate nutrition, mainly in patients with advanced age; therefore, it would be advisable to restrict the ingestion of inorganic P which is totally absorbed and has no real specific benefits. Additionally, one should be aware about the high levels of inorganic P present in additives and many soft drinks [13,14,15,16].

In general, patients are unaware of the amount of inorganic P contained in their diets and therefore it is not easy to instruct patients how to reduce the ingestion of inorganic P and provide information about which compounds should be avoided. In a previous article [17], we reported that careful dietary interviews revealed that the urinary P/UUN ratio was indicative of an excess intake of inorganic P relative to the consumption of proteins. However, it remains unclear whether in a healthy individual, a change in the intake of inorganic P resulted in a commensurate modification in the urine P/UUN ratio. The proteins absorbed are metabolized into U which is excreted in the urine; the organic P derived from protein sources would also be excreted in urine, thus excessive protein intake will increase both P and U in urine and therefore should not increase the urinary P/UUN ratio. However, a relative increase in inorganic P intake should produce an elevation of the P/UUN ratio. Therefore, the urinary P/UUN ratio should allow discrimination of the intake of inorganic versus organic P.

The main objective of our studies was to determine if an increased intake of a known amount of inorganic P translates into a measurable increase in the urine P/UUN ratio of healthy subjects and stage 3 CKD patients.

## 2. Materials and Methods

### 2.1. Study Design

These are longitudinal and prospective studies; the first study is carried out in healthy volunteers and the second study in CKD stage 3 patients. Both were conducted in compliance with the Declaration of Helsinki (1975, revised in 2013), and the protocol received approval from the Ethics Committee of Córdoba (Research Ethics Committee of Córdoba, Spain. First study: registration number: 330; committee reference number: 5276; approval date: 21-DEC-2021. Second study: registration number: SICEIA-2024-000326; committee reference number: 5875; approval date: 2-APR-2024). All subjects signed informed consent for inclusion in the study. No formal sample size calculation was conducted, as this was an exploratory proof-of-concept study. The number of participants was determined pragmatically according to feasibility.

Inclusion criteria: All participants were between 18 and 85 years old with an estimated glomerular filtration rate (eGFR) greater than 60 mL/min/1.73 m^2^ and without albuminuria (in Study-1) and between 30 and 59 mL/min/1.73 m^2^ (in Study-2).

Exclusion criteria: Individuals with autoimmune, intestinal, or cardiovascular diseases were excluded; also excluded were pregnant or breastfeeding women, and those who did not fulfill the inclusion criteria.

Groups design: The Study-1, in healthy volunteers, included 18 subjects. Data collection and testing were conducted during a two-week period. In the first week (visit 1), the participants underwent a 3-day dietary survey to estimate the average daily intake of protein and P as well as the source of P ingested; in addition, during these 3 days of dietary interview, patients were asked to collect 24 h urine at the third day. In the second study, in CKD patients, blood samples were also obtained to evaluate renal function and phosphaturic hormones. In both studies, block randomization was used to equally distribute the subjects in three different groups to be exposed to different diet patterns. In the second week (visit 2), the dietary P was modified as follows. The first group (n = 6), considered the “control” group, were asked not to change their usual diet. The second group (n = 6), called “soda zero”, incorporated into their diet 3 daily cans of soda, 33 cl, zero calories, during the first 3 days (total of 222 mg P of inorganic phosphate). The soda is a product that contains P only in the inorganic form. The third group (n = 6), “processed cheese”, incorporated 230 g/day of processed cheese in their diet. Processed cheese contains additives with inorganic phosphate and has proteins with organic phosphate; the amounts of the two types of phosphate are specified in the dietary information. As in week 1, all participants underwent a dietary survey during the first 3 days, with 24 h urine collection on the third day as in week 1, and blood samples from CKD patients.

### 2.2. Estimation of Protein and P Intake

Protein and P intake were estimated using a table that includes the composition of the foods consumed in the south of Spain. The information on the content of P shown in this table coincides with the values indicated in other sources of food information, the Spanish Database of Food Composition “BEDCA” [18]. In addition, diet composition was also calculated using the diet calibrator [19]. The food composition values collected in this database have been obtained from different sources, including laboratories, the food industry and scientific publications. This database was built according to the European standards developed by the EuroFIR European Network of Excellence and is included in the list of food composition databases of the EuroFIR Association. The dietary records are considered an appropriate method for assessment of dietary intake by the *European Food Safety Authority* (EFSA) pan-European dietary survey. However, the data collected is an estimate based on the self-reported number of processed products eaten by each study volunteer through the dietary survey.

Before the initiation of the study and proceeding with the dietary survey, the volunteers were trained in how to properly describe food, amounts consumed, cooking methods, etc. Food consumption was recorded daily both per meal and between meals, indicating all types of food or drink consumed as well as the intake of processed products. The surveys were reviewed with each volunteer daily to record the intake and add the omitted items and amounts.

### 2.3. Urine Analysis

Twenty-four-hour urines were collected in both weeks. Each individual collected two 24 h urine collection: baseline and intervention (the third day of the week). Urine content of P, creatinine, U, calcium and sodium were measured and the ratios of P/UUN and P/creatinine in urine were obtained. In any individual, the total daily amount of creatinine excreted in urine is constant; in fact, two paired urines collected by each individual were less than 4% suggesting that urines were collected appropriately. To convert urea to urea nitrogen, urinary urea values were divided by 2.14.

### 2.4. Blood Analysis

Blood was collected for measurements of serum biochemistry and complete blood count. Laboratory tests included serum creatinine, U, glucose, uncorrected calcium, P, magnesium, albumin, lipids, ferritin, iron and C reactive protein (CRP). The estimated glomerular filtration rate (eGFR) was calculated using the Chronic Kidney Disease Epidemiology Collaboration (CKD-EPI) formula. Complete blood count was measured with an ABX Pentra 120 Retic^®^ (Horiba, Kyoto, Japan). Human plasma c-terminal FGF23 (c-FGF23) was determined by ELISA (Immutopics, San Clemente, CA, USA). Intact PTH level was quantified by ELISA (Immutopics, San Clemente, CA, USA). Soluble klotho was determined by ELISA (Immutopics, San Clemente, CA, USA).

### 2.5. Statistical Analysis

Block randomization was used to warrant the same number of subjects per group. A descriptive study of the variables was performed, showing the quantitative variables as median (IQR) or as categorical variables in percentage (%). Normality was tested for each variable. Nevertheless, considering the small number of subjects per group, we relied on the recommendations proposed by Siegel [20] and Altman [21] to ensure a more appropriate statistical interpretation. Therefore, the Wilcoxon rank test was performed to compare the means of paired groups. Simple correlation analysis (Spearman correlation test) was used to assess the relationship between numerical variables. In addition, one-way ANOVA and linear mixed models were applied to evaluate group and time interactions.

## 3. Results

### 3.1. Study-1: Healthy Volunteers

The mean age of participants was 40.0 ± 16.2 years (range 24 to 71). A total of 44% (n = 8) were men and all subjects had an eGFR > 60 mL/min/1.73 m^2^ with no abnormalities in urinalysis. High blood pressure was present in two of the volunteers, both well controlled with one medication (no diuretic), and none of the volunteers had diabetes or dyslipidemia. The mean body mass index (BMI) was 28.1 ± 5.0 kg/m^2^. The characteristics of volunteers were similar between the three groups. Characteristics of baseline intake are shown in Table 1 while Table 2 shows the basal biochemical parameters in 24 h urine. The total protein intake collected was higher than that reported in Spain in the ANIBES study (75.0 ± 23.0 g) [22].

The median baseline intake of P analyzed by both methods BEDCA and diet calibrator was 1.5 (1.2–1.7) g/day. The sources of P were the following: animal protein 0.79 (0.69–0.99), BEDCA, 0.77 (0.68–0.97), diet calibrator; vegetable protein 0.45 (0.33–0.66), BEDCA, 0.48 (0.34–0.62), diet calibrator; and inorganic P 0.15 (0.08–0.26), BEDCA and diet calibrator. The total amount of P excreted in the urine in basal conditions was 1004 (749.2–1305) mg, and the mean urinary P/UUN ratio value was 88.4 (72.0–97.0).

Table 3 shows the information on dietary intake analyzed by BEDCA and diet calibrator in the basal situation and after the intervention. Table 3A corresponds to the “control” group (no change in the diet), Table 3B refers to the “zero soda” group (inorganic P was added to the diet in the form of zero soda) and Table 3C shows the data of the “processed cheese” group (organic and inorganic P added to the diet in the form of processed cheese).

According to both dietary databases, during the intervention week the control group showed a statistically significant increase in vegetable protein and total protein intake, without a concomitant rise in phosphate intake.

The total protein intake of the “zero soda” group was similar in both baseline and the intervention week, and there was an increase in the amount of animal protein ingested during the week of intervention. The intake of inorganic P was increased during the intervention. However, the dietary registries could not detect significant differences in this inorganic P intake.

During the intervention, in the group of “processed cheese”, the protein intake increased, but the changes did not reach statistical significance. Also, the intake of animal and inorganic P was increased which is explained by the composition of processed cheese, which has a high content of proteins and a high content of inorganic P.

The parameters measured in urine are shown in Table 4. The data of the “control” group that was instructed not to change their diet is shown in Table 4A. As expected, in this group the urinary excretion of P and the ratio P/UUN did not change. The diuresis after the intervention was lower, although there was no intervention per se since there were no changes in the diet; for this reason, it is not clinically relevant and does not affect the objective of the study. Table 4B refers to the soda zero group (a total of 222 mg inorganic P was added to the diet by drinking three cans of 33 cl of zero-calories soda). The urine P/UUN ratio increased after intervention reflecting the increased intake of inorganic P. The correlation between P and U content in urine in this group is shown in Figure 1; the increased intake of inorganic P moves upward the regression line, and the ingestion of inorganic P increased while the protein ingestion, reflected as U in urine, did not change. Finally, Table 4C shows the data from the processed cheese group. In this group, an increase in calcium excretion was detected after cheese intake, which may be due to the increased calcium intake provided in the intervention. However, although the dietary survey indicated an increase in the intake of inorganic P, an excess inorganic P intake is not reflected as an increase P/UUN ratio in urine in the intervention week with respect to the baseline. This is because the P contained in processed cheese is not only inorganic but also contains a large amount of P from animal protein. Thus, the method used to analyze intake (BEDCA and diet calibrator) is not precise enough to quantify what is the proportion of each type of P contained in foods. In volunteers from this group, the values of urinary P/UUN ratio did not detect a relative increase in the content of inorganic P in the diet. According to the results obtained in the urine, the addition of “processed cheese” to the diet produced an increase in urinary excretion of both P and U; the increase in both P and U were parallel, therefore the ratio P/UUN did not change. The individual changes in the urinary P/UUN ratio in the three groups are shown in Figure 2. It is observed that all but one of the individuals in the soda zero group increased the P/UUN ratio in urine.

### 3.2. Study-2: CKD Patients

The mean age of patients was 39 years (range 36 to 75) and 67% (n = 12) were men. The mean eGFR was 44 ± 13.0 mL/min/1.73 m^2^ without albuminuria. Hypertension was present in 89% of patients (n = 16), insulin resistance in 50% (n = 9) and dyslipidemia in 83% (n = 15). The mean BMI was 33 ± 1.1 kg/m^2^. Characteristics of baseline intake and after intervention are shown in Table 5. In this study, only the diet calibrator was used to estimate the total intake of patients since both methods (BEDCA and diet calibrator) did not show differences in the first study. Total protein and phosphate intake remained stable across both visits in the control and soda zero groups. In contrast, the processed cheese group showed an increase in both protein and phosphate intake after the intervention.

Urine biochemistry per group is shown in Table 6. Notably, despite the higher phosphate intake in both the soda zero and processed cheese groups, only the soda zero group showed a significant increase in urinary phosphate excretion.

In Study-2, which included patients with CKD stage 3, the Wilcoxon test revealed a significant increase in the urinary P/Cr ratio in the soda zero group after the intervention (*p* = 0.028). However, this difference did not reach statistical significance when analyzed using linear mixed-effects models (*p* = 0.099), suggesting that the SODA ZERO group might have concomitantly ingested more animal protein responsible for a larger creatinine production. For the P/UUN, no significant intra-group differences were found using either statistical method.

When comparing between groups, both ANOVA post hoc analyses and linear mixed-effects models identified a statistically significant increase in urinary phosphate excretion (expressed as P/UUN) in the soda zero group compared to the processed cheese group after the dietary intervention. These findings suggest that an acute phosphate load from inorganic sources (e.g., soda) has a more pronounced effect on urinary phosphate handling than organic sources (e.g., cheese), even in individuals with reduced kidney function.

This is different from the healthy cohort, where intra-group comparisons were sufficient to detect changes; in CKD patients the between-group approach (via linear mixed-effects modeling) proved more appropriate to detect differences in phosphate handling. These findings are summarized in Table 7 and illustrated in Figure 3.

Bonferroni-adjusted post hoc comparisons showed a significant difference between the soda zero group (inorganic phosphate) and the processed cheese group (organic phosphate) after the intervention (Visit 2).

In Study-2, unlike Study-1, blood samples in CKD patients were collected on the fourth day. No significant within-group changes were observed in biochemical parameters after the intervention. Blood biochemistry by group is presented in Table 8. A decrease in 25-hydroxyvitamin D and 1,25-dihydroxyvitamin D levels was observed in the soda zero and processed cheese groups, respectively. Additionally, the processed cheese group exhibited a reduction in serum magnesium levels.

To assess whether phosphaturic hormones could reflect short-term dietary phosphate changes, circulating levels of FGF23 and klotho were evaluated. Although a statistically significant reduction in cFGF23 was observed within the soda group using the Wilcoxon test (Table 8), this result was not confirmed in the linear mixed-effects model or in pairwise comparisons between groups suggesting that cFGF23 may show intra-group variability, but lacks robustness as a marker of acute phosphate load in CKD when evaluated using adjusted statistical models (Table 9). Serum iFGF23 and klotho levels (both plasma and urinary) remained stable, and no significant differences were found between the soda and cheese groups (Table 8).

These findings suggest that, in patients with CKD, phosphaturic hormones are less responsive to acute dietary phosphate intake, particularly when compared to urinary markers such as the P/UUN ratio.

## 4. Discussion

The purpose of the present article was to show that the intake of inorganic P relative to the intake of organic P contained in proteins is reflected by the ratio P/UUN in urine. We modified the intake of either inorganic or organic P and analyzed the urine excretion of P and U. The study was performed in healthy volunteers that were randomized into three groups: “controls”, “soda zero” and “processed”. Controls were asked to maintain the same dietary habits. The “soda zero” were asked to drink three cans of 33 mL each daily for three consecutive days to increase inorganic P intake and ensure the rest of their diet remained unchanged. Finally, the “processed cheese” group were instructed to add processed cheese, 230 g daily for 3 days, which should result in an increased intake of both organic and inorganic P. The idea was to demonstrate that the ratio P/UUN was able to identify a high intake of inorganic P relative to the intake of organic P. Our results performed in healthy individuals showed that the measurement of the P/UUN ratio in urine reflects the intake of inorganic P relative to that of organic P while databases and diet calibrators failed to estimate inorganic phosphate content.

The significant data obtained after estimation using BEDCA and diet calibrator did not impact the performance of clinical studies. In this sense, the control group in Study-1 increased total protein intake due to the vegetable protein intake but did not modify phosphate intake; this may be a bias due to participation in a clinical study. On the other hand, the processed cheese group in Study-2 increased both total protein and phosphate intake, after the intervention with organic and inorganic phosphate, but a higher phosphate intake was not estimated in the soda zero group. Although both methods are able to report the diet composition, it fails to estimate the proportion of inorganic phosphate ingested. Therefore, it is advisable to measure P/UUN in urine to assess the intake of inorganic P either in healthy individuals or CKD patients.

Consistent with the protocol, the control group showed no meaningful change in urinary P/UUN across visits. In healthy volunteers, the soda-zero arm—exposed to an exclusively inorganic phosphate load—demonstrated a significant rise in P/UUN, whereas the processed-cheese arm did not; in the latter, urinary phosphate and urea increased in parallel, leaving the ratio essentially unchanged. In CKD stage 3 patients, within-group changes were modest; however, between-group analyses evaluated with ANOVA with post hoc corrections and linear mixed-effects models showed a higher P/UUN after the intervention in the soda-zero group compared with the processed-cheese group. Taken together, these findings indicate that an acute inorganic phosphate load is captured by the urinary P/UUN ratio, while concurrent increases in protein-derived (organic) phosphate do not modify the ratio. Dietary registries (BEDCA and diet calibrator) adequately reflected total protein and phosphate intake but were not reliable for quantifying inorganic phosphate from additives—particularly in beverages—underscoring the utility of urinary P/UUN as a pragmatic biomarker of short-term inorganic phosphate exposure.

Additives containing inorganic P contribute to an excessive intake of P which is easily absorbed at the intestinal level. There is increasing evidence of an association between high P intake and adverse health effects in otherwise healthy populations [23]. The magnitude of potential negative effects of abuse of inorganic P additives remains to be determined. It has been known for years that the intake of inorganic P may lead to excessive P exposure, toxicity and cardiovascular disease in the general population [24]. It is straightforward to determine accurately the dietary content of inorganic P; an initial step should be to determine with acceptable precision the exposure to inorganic P additives in the food supply [25]. The results of the present study show that the determination of the P/UUN ratio in urine is a valid and simple method to assess the dietary ingestion of inorganic P.

The median basal intake of P in the study volunteers was 1.5 (1.2–1.7) g/day. In the Spanish population, it has been reported that the average daily intake of P is 1.3 g [26]. In a situation of P balance, the excretion of P in 24 h urine collection should reflect the P intake [22], although it should be considered that the circadian pattern of serum P is modifiable by P intake [27]. Several authors have argued that phosphaturia measured in a 24 h urine collection does not correlate with the total calculated dietary P [28,29,30]. These findings are not surprising since phosphaturia reflects the intestinal absorption of P, which varies according to the source of P ingested. The EFSA reports that the range of P intake should be between 1000 and 1767 mg/day. However, as previously mentioned, the amount of P absorbed by the intestine varies according to the source of P ingested; the inorganic P is almost completely absorbed, but in contrast, the intestinal absorption of organic P is variable and dependent on the type proteins [31]. Moreover, the amount of inorganic P contained in food additives significantly masks the relationship between calculated P intake and P excretion in urine. Thus, P additives represent a significant and “hidden” P load in modern diets [16,32].

The KDOQI (*Kidney Disease Outcomes Quality Initiative*) 2020 clinical practice guideline for nutrition in CKD suggested reducing protein intake to 0.8 g/kg/day in patients with eGFR < 30 mL/min/1.73 m^2^ [33]. The level of eGFR appears to be affected by protein intake [31], healthy adults with lower protein intake may have lower mean eGFR but usually it is not below 60 mL/min/1.73 m^2^ [34]. It is widely accepted that the value of 24 h U excretion is a measure of the protein absorbed; likewise urinary P reflects the P that has been absorbed. It may be advisable to determine to what extent a favorable effect of a moderate protein restriction is reliant on the change in both urinary P and U. There are reports indicating that a progressive increase in FGF23 predicts a rapid progression of kidney disease and mortality [35]; the increase in FGF23 is an adaptative response to the phosphate load that is related to a high intestinal absorption of P relative to reduced capacity of renal filtration of P. Patients with CKD should control the intake of P to prevent not only renal damage but also the elevation of FGF23, a hormone that is also associated with cardiovascular morbidity and mortality. It is important to educate patients to select foods without P additives [36]. Currently, there is no specific recommendation in relation to restrictions on inorganic P, which is easily absorbable and lacks nutritional value.

To further explore the utility of this ratio in a clinical population, we extended the protocol to patients with CKD stage 3. The P/UUN ratio, adapted for this setting, similarly showed a significant increase only in the group exposed to inorganic phosphate (soda group), with no significant changes in the control or processed cheese groups. These results confirm that the P/UUN ratio remains a valid biomarker even in individuals with moderate renal dysfunction.

Importantly, phosphaturic hormones such as FGF23 and klotho were also measured to determine whether they could reflect acute phosphate intake. However, neither serum FGF23 nor plasma klotho showed significant intra-group changes. Although a between-group difference was observed between the soda and cheese arms in linear mixed models, this was not replicated within groups or supported by consistent trends. These results are consistent with the known physiology of FGF23 and klotho, which regulate phosphate homeostasis over longer time frames and in response to sustained phosphate retention, rather than acute dietary variations.

A main limitation of the study is the short number of subjects in each group and that no formal sample size calculation was performed. This study was conceived as an exploratory proof-of-concept trial intended to assess the feasibility of using the urinary P/UUN ratio to detect short-term changes in inorganic phosphate intake. While the small number of participants restricts generalizability, the consistent detection of differences despite this limitation supports the validity of the approach and provides a rationale for future studies with larger, adequately powered cohorts. It must be taken into consideration that urine was carefully collected with a high degree of precision as reflected by minimal variation in the total amount of creatinine collected in the same volunteer before and after the intervention. Also, we may argue that each participant underwent a replicated dietary survey for three days to account for protein and P intake at two different times. At the beginning of the second week of follow-up, we used block randomization as described elsewhere to maximize statistical power given the limited sample size [37]. Further, replicated dietary surveys were repeated. Additionally, our approach of repeated measurements may also increase statistical power for detecting changes in small-sample-size studies [38]. Thus, at the end of each period, end of week 1 and at the end of week 2, 24 h urine samples were collected. Thus, we collected a total of 72 urine samples from 24 h collections. As a proof of concept, we previously showed [17] that the urinary P/UUN ratio accurately identified those CKD subjects with larger inorganic P consumption by means of having eaten processed food. Thus, repeated measures and the determination of the P/UUN ratio may overcome a limited sample to detect the ingestion of inorganic phosphate.

A second limitation of the present study is the absence of blinding of participants and assessors. Given the nature of the intervention, which required the incorporation of specific and easily recognizable food items (soda or processed cheese), blinding was not feasible. To reduce the potential impact of this limitation, standardized instructions were provided to all participants, dietary records were closely reviewed by the investigators and objective biochemical measurements in urine were used as the primary outcome. These measures were intended to minimize the risk of bias despite the lack of blinding.

Another limitation of this study is the short duration of the intervention (3 days). This time frame was intentionally chosen to evaluate whether acute changes in dietary inorganic phosphate intake could be detected through the urinary P/UUN ratio. While this design does not allow conclusions regarding medium- or long-term effects, particularly in patients with CKD, it served to demonstrate the feasibility of the approach and to provide preliminary evidence that justifies future studies with longer follow-up and larger cohorts.

An additional limitation of the study is that we did not obtain blood samples to analyze changes in P and U following the different interventions in the healthy population. Being volunteers with normal renal function, one might expect that the interventions performed should not significantly alter the serum values of these parameters, although honestly, we cannot be certain since these parameters were not measured. In any case, the P/UUN ratio in urine has been sufficient to discern the amount of inorganic P ingested.

## 5. Conclusions

The urinary P/UUN ratio emerges as a potential tool to detect excessive inorganic phosphate intake, but validation in larger and more diverse CKD cohorts is required before clinical implementation.

## Figures and Tables

**Figure 1 nutrients-17-03323-f001:**
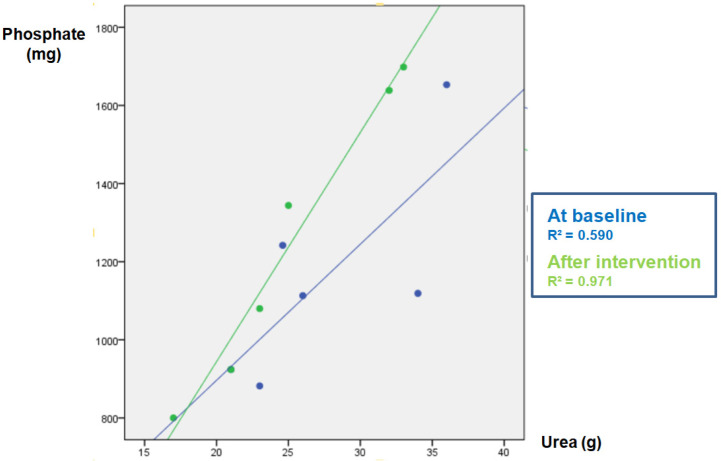
Study-1: Phosphate (P) vs. urea (U) correlation in baseline urine and after intervention in the SODA ZERO group.

**Figure 2 nutrients-17-03323-f002:**
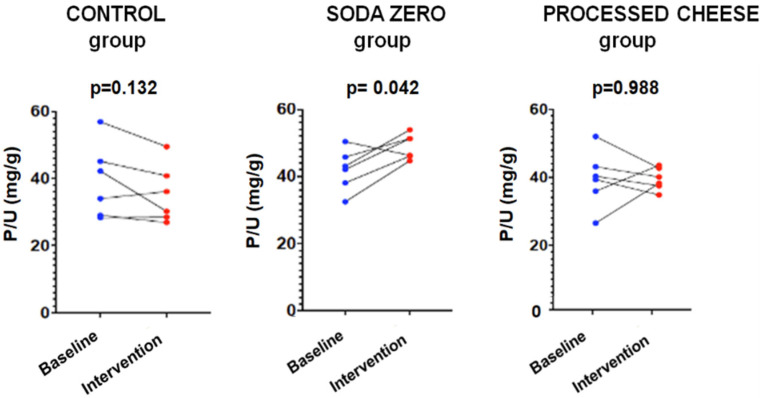
Study-1: P/UUN at baseline and after intervention in each group.

**Figure 3 nutrients-17-03323-f003:**
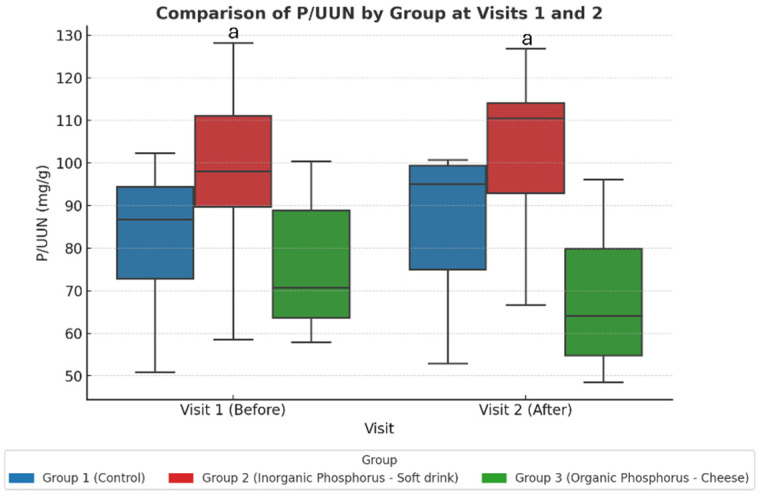
Study-2: The urine P/UUN ratio per group at baseline (visit 1) and after dietary intervention (visit 2). When comparing the three groups across the two visits, an increase in the P/UUN ratio is observed in the group consuming inorganic phosphate compared to the group consuming organic phosphate. Values represent adjusted marginal means estimated from linear mixed-effects models. Statistical significance was assessed using ANOVA post hoc and linear mixed-effects models. ^a^ *p* < 0.05 Group 2 vs. Group 3.

**Table 1 nutrients-17-03323-t001:** Study-1: Baseline intake data analyzed using Spanish Food Composition Database (BEDCA) and diet calibrator (n = 18).

	BEDCA	Median (IQR)
Proteins (g/day)	Total	108.0 (86.5–136.1)
	Animal	80.0 (58.8–92.0)
	Vegetables	29.7 (18.0–37.1)
Phosphate (g/day)	Total	1.5 (1.2–1.7)
	Organic animal	0.79 (0.69–0.99)
	Organic vegetable	0.45 (0.33–0.66)
	Inorganic	0.15 (0.08–0.26)
	**DIET CALIBRATOR**	**Median (IQR)**
Proteins (g/day)	Total	104.1 (86.6–133.8)
	Animal	78.2 (58.7–90.1)
	Vegetables	29.5 (17.9–37.0)
Phosphate (g/day)	Total	1.5 (1.2–1.7)
	Organic animal	0.77 (0.68–0.97)
	Organic vegetable	0.48 (0.34–0.62)
	Inorganic	0.15 (0.08–0.26)

**Table 2 nutrients-17-03323-t002:** Study-1: Baseline biochemical parameters in urine 24 h. (n = 18).

Urinary Parameters	Median (IQR)
Phosphate (P)(mg)	1004 (749.2–1305)
Creatinine (Cr)(g)	1.2 (1.0–2.0)
Urea (U) (g)	24.3 (20.4–34.5)
P/Cr (mg/g)	719.4 (567.4–795.6)
P/UUN (mg/g)	88.4 (72.0–97.0)
U/Cr (mg/g)	18.1 (14.2–21.0)
Sodium (Na) (mEq)	202.7 (151.4–286.4)
Na/Cr (mEq/g)	148.3 (114.7–200.7)
Calcium (Ca) (mg)	135.5 (88.1–191.8)
Ca/Cr (mg/g)	86.3 (65.3–116.9)
Diuresis 24 h (L)	2.2 (1.5–2.9)

**Table 3 nutrients-17-03323-t003:** Study-1: Intake data analyzed using the Database Spanish Food Composition Calibrator (BEDCA) and diet calibrator. Baseline and intervention. Statistically significant values are highlighted in bold.

(A) CONTROL Group (n = 6)	BEDCA	Baseline	Intervention	*p*
Proteins (g/day)	Total	74.8 (62.1–94.7)	93.7 (72.6–113.6)	**0.028**
	Animal	53.5 (43.6–67.2)	62.4 (49.7–77.1)	0.116
	Vegetables	21.9 (14.5–30.1)	30.4 (22.6–36.2)	**0.028**
Phosphate (g/day)	Total	1.2 (1.0–1.4)	1.2 (1.0–1.6)	0.600
	Organic animal	0.67 (0.62–0.74)	0.62 (0.46–0.84)	0.463
	Organic vegetable	0.39 (0.23–0.48)	0.46 (0.40–0.48)	0.075
	Inorganic	0.16 (0.06–0.32)	0.15 (0.12–0.28)	0.600
	**DIET CALIBRATOR**	**Baseline**	**Intervention**	
Proteins (g/day)	Total	74.8 (62.1–93.1)	93.7 (72.6–113.6)	**0.028**
	Animal	53.5 (43.6–66.4)	62.4 (49.7–77.0)	0.116
	Vegetables	20.4 (14.6–30.1)	30.4 (22.6–36.2)	**0.028**
Phosphate (g/day)	Total	1.2 (1.0–1.4)	1.3 (1.0–1.6)	0.600
	Organic animal	0.66 (0.63–0.74)	0.62 (0.46–0.84)	0.463
	Organic vegetable	0.36 (0.23–0.48)	0.46 (0.40–0.48)	0.075
	Inorganic	0.16 (0.06–0.33)	0.15 (0.12–0.28)	0.600
**(B) SODA ZERO Group (n = 6)**	**BEDCA**	**Baseline**	**Intervention**	** *p* **
Proteins (g/day)	Total	120.7 (98.6–151.2)	120.5 (99.5–137.1)	0.463
	Animal	77.1 (65.0–124.5)	65.2 (49.8–80.8)	**0.028**
	Vegetables	30.8 (18.8–50.2)	52.2 (46.0–58.0)	0.116
Phosphate (g/day)	Total	1.7 (1.3–1.8)	1.8 (1.3–2.0)	0.917
	Organic animal	0.83 (0.75–1.25)	0.86 (0.59–0.93)	0.075
	Organic vegetable	0.45 (0.33–0.75)	0.77 (0.47–0.82)	0.249
	Inorganic	0.14 (0.05–0.22)	0.15 (0.10–0.33)	0.345
	**DIET CALIBRATOR**	**Baseline**	**Intervention**	** *p* **
Proteins (g/day)	Total	120.0 (98.4–151.0)	120.5 (99.5–137.1)	0.463
	Animal	77.1 (64.8–124.5)	65.2 (49.8–80.8)	**0.028**
	Vegetables	30.5 (18.7–49.7)	52.2 (46.0–57.9)	0.116
Phosphate (g/day)	Total	1.6 (1.3–1.8)	1.8 (1.3–2.0)	0.600
	Organic animal	0.82 (0.74–1.11)	0.86 (0.59–0.93)	0.075
	Organic vegetable	0.50 (0.36–0.72)	0.77 (0.47–0.82)	0.249
	Inorganic	0.14 (0.05–0.22)	0.15 (0.10–0.33)	0.345
**(C) PROCESSED CHEESE Group** **(n = 6)**	**BEDCA**	**Baseline**	**Intervention**	** *p* **
Proteins (g/day)	Total	122.9 (111.2–142.2)	155.7(126.0–175.1)	0.116
	Animal	87.7 (85.7–102.8)	110.2 (92.1–123.3)	0.173
	Vegetables	35.5 (23.1–42.9)	36.6 (26.1–68.2)	0.249
Phosphate (g/day)	Total	1.7 (1.4–2.1)	1.8 (1.6–2.1)	0.249
	Organic animal	0.91 (0.77–1.23)	0.73 (0.59–0.80)	**0.046**
	Organic vegetable	0.66 (0.35–0.70)	0.62 (0.53–0.83)	0.173
	Inorganic	0.12 (0.09–0.36)	0.45 (0.40–0.51)	**0.028**
	**DIET CALIBRATOR**	**Baseline**	**Intervention**	** *p* **
Proteins (g/day)	Total	122.7 (105.1–139.0)	155.7 (126.0–175.1)	0.075
	Animal	86.7 (82.8–101.1)	110.2 (92.1–123.3)	0.173
	Vegetables	34.5 (22.3–42.7)	36.6 (26.1–60.7)	0.173
Phosphate (g/day)	Total	1.7 (1.3–2.1)	1.8 (1.6–2.1)	0.249
	Organic animal	0.88 (0.76–1.21)	0.74 (0.58–0.80)	0.075
	Organic vegetable	0.63 (0.36–0.72)	0.62 (0.53–0.77)	0.249
	Inorganic	0.12 (0.09–0.35)	0.45 (0.40–0.51)	**0.028**

**Table 4 nutrients-17-03323-t004:** Study-1: Biochemical parameters in 24 h urine parameters at baseline and after dietary intervention. Statistically significant values are highlighted in bold.

(A) CONTROL Group (n = 6)	Baseline	Intervention	*p*
Phosphate (P) (mg)	739.5 (517.9–1314)	619.3 (484.6–1399)	0.345
Creatinine (Cr) (g)	1.1 (0.9–2.0)	1.1 (0.8–2.0)	0.345
Urea (U) (g)	20.7 (16.5–28.9)	21.0 (15.7–32.5)	0.345
P/Cr (mg/g)	611.5 (567.2–710.2)	601.6 (539.5–700.7)	0.463
P/UUN (mg/g)	81.6 (61.7–102.9)	71.1 (60.2–92.0)	0.173
U/Cr (mg/g)	15.7 (13.3–20.7)	17.0 (14.7–21.8)	0.463
Sodium (Na) (mEq)	224.6 (163.4–253.8)	132.5 (104.6–195.9)	**0.028**
Na/Cr (mEq/g)	138.9 (118.1–243.2)	98.3 (88.5–142.2)	**0.046**
Calcium (Ca) (mg)	149.5 (70.9–286.3)	123.3 (69.4–221.3)	0.345
Ca/Cr (mg/g)	89.1 (69.9–190.6)	106.0 (44.6–149.4)	0.116
Diuresis 24 h (L)	3.0 (2.5–3.4)	2.1 (1.7–2.7)	**0.046**
**(B) SODA ZERO Group (n = 6)**	**Baseline**	**Intervention**	
Phosphate (P) (mg)	1115 (913.5–1345)	1212 (893.0–1654)	0.917
Creatinine (Cr) (g)	2.1 (1.2–2.2)	2.0 (1.1–2.5)	0.463
Urea (U) (mg)	25.5 (22.7–34.8)	24.2 (19.8–32.2)	0.463
P/Cr (mg/g)	656.0 (548.4–765.8)	742.0 (564.3–789.9)	0.917
P/UUN (mg/g)	91.5 (78.6–100.5)	104.5 (98.0–111.2)	**0.046**
U/Cr (mg/g)	16.1 (12.9–18.4)	14.2 (11.2–17.1)	0.116
Sodium (Na) (mEq)	177.5 (132.2–333.7)	199.2 (146.0–256.2)	0.6
Na/Cr (mEq/g)	107.3 (60.8–192.5)	111.8 (82.3–162.2)	0.917
Calcium (Ca) (mg)	148.7 (113.6–181.1)	131.5 (92.0–264.3)	0.6
Ca/Cr (mg/g)	77.2 (56.8–101.6)	87.0 (54.7–128.0)	0.463
Diuresis 24 h (L)	1.6 (1.0–2.2)	1.5 (1.0–2.2)	0.5
**(C) PROCESSED CHEESE** **(n = 6)**	**Baseline**	**Intervention**	
Phosphate (P) (mg)	1061 (731.4–1498)	1250 (933.0–1500)	0.6
Creatinine (Cr) (g)	1.1 (0.9–1.9)	1.4 (1.1–2.1)	0.249
Urea (U) (mg)	24.2 (20.4–35.4)	30.6 (25.7–38.3)	0.249
P/Cr (mg/g)	851.8 (689.7–1001)	779.8 (630.4–992.0)	0.917
P/UUN (mg/g)	85.3 (71.9–97.0)	83.9 (79.0–91.9)	0.917
U/Cr (mg/g)	20.3 (18.7–23.8)	19.3 (16.0–26.8)	0.6
Sodium (Na) (mEq)	182.3 (159.9–352.6)	261.6 (200.2–290.9)	0.917
Na/Cr (mEq/g)	176.5 (161.4–200.7)	139.1 (118.2–259.6)	0.753
Calcium (Ca) (mg)	130.0 (78.1–176.3)	201.0 (153.0–271.0)	**0.028**
Ca/Cr (mg/g)	89.0 (62.4–158.9)	117.2 (79.4–259.1)	**0.028**
Diuresis 24 h (L)	2.0 (1.3–2.7)	2.3 (1.3–2.6)	0.916

**Table 5 nutrients-17-03323-t005:** Study-2: Intake data analyzed using the diet calibrator. Baseline and intervention. Statistically significant values are highlighted in bold.

(A) CONTROL Group (n = 6)	Baseline	Intervention	*p*
Proteins (g/day)	68.6 (44.6–92.1)	73.6 (46.4–107.5)	0.116
Phosphate (g/day)	1.2 (0.9–1.3)	1.3 (0.9–1.3)	0.345
**(B) SODA ZERO Group (n = 6)**	**Baseline**	**Intervention**	** *p* **
Proteins (g/day)	72.4 (57.2–94.8)	74.4 (57.2–105.1)	0.463
Phosphate (g/day)	1.2 (0.97–1.32)	1.3 (0.91–1.5)	0.6
**(C) PROCESSED CHEESE Group (n = 6)**	**Baseline**	**Intervention**	** *p* **
Proteins (g/day)	96.2 (69–99.8)	114.2 (95.1)	**0.046**
Phosphate (g/day)	1.5 (1.1–1.6)	2.3 (2.1–2.7)	**0.028**

**Table 6 nutrients-17-03323-t006:** Study-2: Biochemical parameters in 24 h urine at baseline and after dietary intervention. Statistically significant values are highlighted in bold. P: phosphate, Cr: creatinine, UUN: urine urea nitrogen.

(A) CONTROL Group (n = 6)	Baseline	Intervention	*p*
P/Cr (mg/g)	590.0 (515.1–682.5)	605.1 (585.0–767.5)	0.249
P/UUN (mg/g)	86.8 (64.8–97.5)	95.1 (64.6–100.8)	0.345
Diuresis 24 h (L)	2.4 (1.5–3.3)	2.5 (1.7–3)	0.173
**(B) SODA ZERO Group (n = 6)**	**Baseline**	**Intervention**	
P/Cr (mg/g)	720.1 (512.5–912.5)	870.0 (662.5–1130)	**0.028**
P/UUN (mg/g)	98.1 (81.1–117.6)	110.6 (82.9–117)	0.6
Diuresis 24 h (L)	2.1 (2–2.5)	2.3 (2.1–2.8)	0.273
**(C) PROCESSED CHEESE Group (n = 6)**	**Baseline**	**Intervention**	
P/Cr (mg/g)	545.2 (402.5–697.5)	605.1 (375–735)	0.345
P/UUN (mg/g)	70.8 (61.1–95.6)	64.4 (53–85.9)	0.249
Diuresis 24 h (L)	2.1 (2–2.5)	2.3 (1.1–3.2)	0.343

**Table 7 nutrients-17-03323-t007:** Adjusted change in marginal means of urinary phosphate-to-urea nitrogen ratio (P/UUN, mg/g) by group and visit, estimated from linear mixed-effects models. Statistically significant values are highlighted in bold.

COMPARISON	Δ P/UUN (mg/g)	95% CI (mg/g)	*p*
Group 1: Visit 2 vs. Visit 1	+3.52	−11.76 to +18.81	0.652
Group 2: Visit 2 vs. Visit 1	+16.46	−11.29 to +44.22	0.393
Group 3: Visit 2 vs. Visit 1	−11.61	−39.36 to +16.15	0.833
Group 2 vs. Group 3 (Visit 2)	**+28.07**	**+0.32 to** +**55.82**	**0.047**

**Table 8 nutrients-17-03323-t008:** Study-2: Biochemical serum parameters at baseline and after dietary intervention. Statistically significant values are highlighted in bold. PTH: parathyroid hormone, i-FGF23: intact fibroblast growth factor, c-FGF23: fibroblast growth factor c-terminal.

(A) CONTROL Group (n = 6)	Baseline	Intervention	*p*
Creatinine (mmol/L)	0.18 (0.15–0.26)	0.16 (0.15–0.26)	0.753
Urea (mmol/L)	14.9 (10.4–17.7)	15.3 (9.6–18.5)	0.752
Glucose (mmol/L)	5.8 (4.86–9.3)	7.1 (5.9–9.6)	0.08
Sodium (mEq/L)	138.0 (135.3–143)	137.0 (136.8–143)	0.498
Potassium (mEq/L)	4.4 (4.1–5)	4.5 (4.3–4.9)	0.596
Chloride (mEq/L)	108.0 (106–110)	109.1 (106–110)	0.194
Calcium (mmol/L)	2.4 (2.1–2.4)	2.3 (2.3–2.3)	1
Phosphate (mmol/L)	1.3 (1.1–1.4)	1.3 (1.1–1.4)	0.892
Magnesium (mmol/L)	0.78 (0.69–0.88)	0.74 (0.71–0.9)	0.917
Albumin (µmol/L)	687.1 (672.1–698.4)	672.0 (630.4–683.3)	0.066
PTH (pg/mL)	130.0 (70.2–318.9)	119.1 (58.9–358.0)	0.917
1,25 hydroxyvitamin D (pg/mL)	29.1 (21.8–45.0)	28.0 (20.8–43.5)	0.238
25 hydroxyvitamin D (pg/mL)	19.6 (15.9–30)	23.8 (18.4–32.8)	0.892
i-FGF23 (pg/mL)	99.8 (63.4–164.1)	113 (68.9–164.2)	0.640
c-FGF23 (RU/mL)	120.0 (75.3–256.1)	122.0 (76.6–244)	0.990
Klotho (pg/mL)	610.0 (172.1–812.0)	463.0 (183.1–729.9)	0.380
**(B) SODA ZERO Group (n = 6)**	**Baseline**	**Intervention**	
Creatinine (mmol/L)	0.16 (0.13–0.20)	0.17 (0.13–0.20)	0.753
Urea (mmol/L)	11.9 (9.2–14.2)	11.5 (10.2–15)	0.248
Glucose (mmol/L)	6.3 (5.2–7.9)	6.6 (4.9–10.1)	0.463
Sodium (mEq/L)	141.1 (138.0–141.3)	138.1 (135.8–142)	0.334
Potassium (mEq/L)	4.7 (4.5–4.7)	4.8 (4.2–5.1)	0.684
Chloride (mEq/L)	108.0 (104.9–109.1)	110.0 (106.8–1101)	0.34
Calcium (mmol/L)	2.3 (2.3–2.4)	2.4 (2.3–2.42)	0.917
Phosphate (mmol/L)	1.1 (0.9–1.3)	1 (0.9–1.3)	1
Magnesium (mmol/L)	0.8 (0.8–0.9)	0.8 (0.8–0.9)	0.686
Albumin (µmol/L)	664.4 (656.9–724.8)	672 (645.6–702.2)	0.683
PTH (pg/mL)	151.9 (96.6–205.6)	167 (78–192)	0.917
1,25 hydroxyvitamin D (pg/mL)	29.5 (21.3–45.0)	31.5 (20.3–49.0)	0.752
25 hydroxyvitamin D (pg/mL)	24.8 (16–37.5)	22.6 (14.9–31.7)	**0.046**
i-FGF23 (pg/mL)	109.6 (67.8–151.4)	98.0 (61.9–158.7)	0.790
c-FGF23 (RU/mL)	131.0 (85.3–159.0)	129.0 (114.0–192.0)	0.160
Klotho (pg/mL)	597.0 (469.0–738.0)	580.0 (491.0–783.0)	0.990
**(C) PROCESSED CHEESE** **(n = 6)**	**Baseline**	**Intervention**	
Creatinine (mmol/L)	0.11 (0.09–0.15)	0.11 (0.09–0.13)	0.173
Urea (mmol/L)	8.0 (6.3–9.8)	8.3 (6.6–10.9)	0.917
Glucose (mmol/L)	5.1 (4.7–5.6)	5.5 (5–6.1)	0.093
Sodium (mEq/L)	142.1 (140.5–148.5)	142.5 (136.8–148)	0.344
Potassium (mEq/L)	4.3 (4.2–4.5)	4.4 (4.3–4.8)	0.244
Chloride (mEq/L)	108.0 (107.3–112.8)	118.5 (105.5–113.8)	0.829
Calcium (mmol/L)	2.4 (2.4–2.5)	2.4 (2.4–2.6)	0.892
Phosphate (mmol/L)	1.2 (1–1.3)	1.3 (1–1.4)	0.498
Magnesium (mmol/L)	1.0 (0.9–1.1)	0.8 (0.7–0.9)	**0.028**
Albumin (µmol/L)	702.2 (679.5–724.8)	694.6 (679.5–713.5)	0.48
PTH (pg/mL)	60.1 (26–127.1)	55.1 (41.1–152.6)	0.463
1,25 hydroxyvitamin D (pg/mL)	43.5 (30.8–56)	36.0 (25.2–51.5)	**0.027**
25 hydroxyvitamin D (pg/mL)	35.0 (20.9–42.6)	32.3 (23.9–49.5)	0.345
i-FGF23 (pg/mL)	79.0 (64.5–106.1)	79.1 (61.3–150.2)	0.590
c-FGF23 (RU/mL)	98.5 (80.0–269.0)	133.0 (106.0–1389)	**0.030**
Klotho (pg/mL)	550.0 (460.0–846.0)	463.0 (379.0–789.0)	0.190

**Table 9 nutrients-17-03323-t009:** Study-2: Comparison of phosphaturic hormone levels between the soda zero group (Group 2) and the processed cheese group (Group 3) after the dietary intervention (Visit 2). Values represent absolute differences in means and Bonferroni-adjusted *p*-values for multiple comparisons. No statistically significant differences were observed between the two groups in any of the evaluated biomarkers.

COMPARISON	Δ Absolute	*p*
Group 2 vs. Group 3 (Visit 2) c-FGF23 (RU/mL)	−538.47	0.463
Group 2 vs. Group 3 (Visit 2) i-FGF23 (pg/mL)	+17.71	1
Group 2 vs. Group 3 (Visit 2) Klotho (pg/mL)	+64.57	1

## Data Availability

The data underlying this article are openly available in the article. No new data were generated or analyzed in support of this research.

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
