# Peer review of "The Phosphate/Urea Nitrogen Ratio in Urine—A Method to Assess the Relative Intake of Inorganic Phosphate"

_nutrients, 2025, doi:10.3390/nu17213323_

Round 1

Reviewer 1 Report

Comments and Suggestions for Authors

The topic is very relevant and important. The authors clearly presented the topic in the Introduction part. Methods are appropriately described. The number of participants per group is low, but the authors clearly stated this as limitation in the Discussion part. Also, the authors clearly stated the second limitation that blood parameters were not used at the beginning of the study for healthy participants. The discussion is focused of the main findings and appropriately citated. Some parts in the text are highlighted in yellow, I do not know way. If this is detection of plagiarism it should be checked, especially in the Result and Discussion part. However, overall, the study is sound and interesting. 

Author Response

Response to Reviewers

We thank the reviewers and editors for their constructive and valuable comments. Below we provide a detailed response to each point raised, indicating how we have revised the manuscript accordingly.

Reviewer 1

We sincerely thank Reviewer 1 for the positive evaluation of our work and for the constructive comments provided. We greatly appreciate the recognition of the study’s relevance, clarity in methods and discussion, and the identification of limitations. We also acknowledge the observation regarding the highlighted text.

We apologize for this formatting issue. The yellow highlighting was unintentional and has been removed.

Reviewer 2 Report

Comments and Suggestions for Authors

I reviewed with interest this proof-of-concept, randomized dietary intervention in 18 healthy volunteers and 18 stage 3 CKD patients, which tested whether the urinary phosphate/urea (P/U) ratio can detect short-term increases in inorganic phosphate intake. The P/U ratio rose significantly only in participants consuming phosphate-containing soda, supporting its potential as a biomarker for inorganic phosphate exposure.

The clinical question addresses a genuine gap in nephrology and nutrition practice. For clinicians, especially nephrologists managing CKD progression, a reliable monitoring tool would be valuable for dietary counseling and public health interventions. 

The design is logical for isolating the effects of different phosphate sources. Inclusion/exclusion criteria are clear. Dietary intake was assessed via validated databases. However, several limitations reduce the robustness:

  • Sample size is extremely small (n=6 per arm), limiting generalizability and statistical power.
  • No blinding of participants or assessors.
  • Dietary assessment tools could not quantify inorganic phosphate reliably, which is central to the study’s premise. 
  • P/U ratio increased significantly only in the soda group, and is consistent across both healthy and CKD cohorts (p=0.042 and p=0.018, respectively). The lack of change in the processed cheese group is biologically plausible given the parallel increase in both inorganic and organic phosphate.
  • Some statistically significant findings (e.g., p=0.042) are marginal and vulnerable to type I error in small samples.
  • The discussion correctly positions the P/U ratio as a potential simple clinical tool while openly acknowledging sample size and duration limitations. However, the authors extrapolate to “daily clinical practice” somewhat prematurely; validation in larger, diverse CKD populations and over longer follow-up is essential before implementation.
  • Justify power calculation or explicitly state the study is exploratory without formal power analysis.
  • Consider including at least baseline and post-intervention serum phosphate, calcium, and FGF23 to corroborate urinary changes.
  • Several sentences require editing for clarity and conciseness (e.g., “heatlhy” → “healthy”).
  • Standardize “P/U ratio” and “phosphate/urea ratio” throughout the manuscript.
  • Provide exact p-values for all non-significant results rather than “n.s.”.

Author Response

Response to Reviewers

We thank the reviewers and editors for their constructive and valuable comments. Below we provide a detailed response to each point raised, indicating how we have revised the manuscript accordingly.

Reviewer 2

We thank Reviewer 2 for the encouraging initial remarks emphasizing the clinical importance of the research question and the value of exploring the urinary P/U ratio as a potential biomarker.

With respect to the comment relative to “Sample size is extremely small (n=6 per arm)” we agree. This was a pilot proof-of-concept study. We now explicitly state in the Methods that no formal power calculation was performed (page 3, line 102), and in the Discussion (page 17, line 476) we highlight that the study is exploratory and hypothesis-generating.

About the comment “No blinding of participants or assessors”, we must accept that due to the nature of the dietary intervention (recognizable foods and beverages), blinding was not feasible. We have added this explicitly to the Limitations section (page 17, line 498).

Regarding to the comment “Dietary assessment tools could not quantify inorganic phosphate reliably”. We agree, in fact, this was a key motivation for testing the urinary P/U ratio as an alternative biomarker due to this parameter is not available and it is important to know it due this type of phosphate is fully adsorbable.

On the other side, we agree with the comment of Reviewer 2 “The P/U ratio increased significantly only in the soda group, consistent across both cohorts (p=0.042 and p=0.018, respectively). The lack of change in the processed cheese group is biologically plausible given the parallel increase in both inorganic and organic phosphate.”

We would like to inform the reviewer that, following their suggestion to report exact p-values instead of “n.s.,” we identified some errors in our initial statistical analyses. These have now been corrected, and the revised manuscript includes the updated results. The overall findings remain consistent with the original version, but are now supported by a more robust statistical analysis.

Additionally, to match with our previous paper, we also refers to the P/U ratio as P/UUN. By using the same term we pretend the reader not to misunderstood the term and to make the term generalizable. Conceptually, P/U and P/UUN ratio are the same, but the use of P/UUN may be more internationally sound.

We thank the reviewer for this insightful interpretation this information is included into the Results (page 6. Line 213) and Discussion (page 16, line 399).

With respect to the comment “Some statistically significant findings are marginal (e.g., p=0.042)” we agree that the small sample size and borderline p-values warrant caution in interpretation. However, we would also like to emphasize that the detection of statistically significant changes despite the limited number of participants and the heterogeneity of the cohorts may actually reinforce the robustness of the association. The fact that the P/UUN ratio consistently increased in the soda group across both healthy and CKD participants suggests that the signal is sufficiently strong to be observed even under suboptimal statistical conditions. This strengthens the rationale for validating the P/UUN ratio in larger, more diverse cohorts.

About the comment “The discussion extrapolates prematurely to daily clinical practice” we thank this suggestion, and we have revised the concluding remarks to tone down extrapolation. The manuscript now frames P/UUN ratio as a “potential tool” requiring validation in larger, diverse CKD cohorts before clinical adoption.

The comment “Justify power calculation or state exploratory design” has been commented in our first response related to Sample size. It has been stated in the Methods that no formal power calculation was performed due this study is exploratory and hypothesis-generating, (page 3, line 102 and page 17, line 476).

Regarding the comment “Consider including baseline and post-intervention serum phosphate, calcium, and FGF23 in healthy volunteers” we acknowledge this

important point. However, as the healthy volunteers had preserved renal function, we did not expect meaningful changes in circulating phosphate, calcium, or FGF23 within such a short intervention period. FGF23, in particular, is known to respond primarily to sustained phosphate retention rather than to acute dietary fluctuations. For this reason, and to minimize unnecessary invasiveness in healthy individuals, we did not obtain blood samples in this group. We recognize this as a limitation and now emphasize it more clearly in the Discussion, noting that future studies in healthy participants should include serum parameters for completeness (page 17 and line 512).

Several sentences require editing for clarity (e.g., “heatlhy by healthy”).

We thank this comment, we have carefully revised the manuscript to correct typographical errors and improve clarity and conciseness throughout.

About the comment “Standardize “P/U ratio and “phosphate/urea ratio” terminology”, we thank the comment and in the revised version of our manuscript we have standardized terminology to “P/U ratio” throughout the text, figures, and tables. In fact, we want to highlight again that P/U ratio has been modified to P/UUN being this one “urinary P to urinary urea nitrogen”. This change was made to maintain the same terminology of previous works from our group

(Nutrients. 2021;13(2):292. doi: 10.3390/nu13020292). As delined before, conceptually, the P/U ratio match with P/UUN as both quatify the same variables but expressed in more internationally accepted terms. To convert the P/UUN ratio to P/U, urinary urea should be multiplied by 2,14.

Finally, respect to the comment “Provide exact p-values for all non-significant results rather than “n.s.”. The statistical analysis has been repeated. We have revised all tables and results to include exact p-values. We have verified that in the original manuscript we made some errors which have been corrected. Among them, we have obtained statistically significant results that were previously not available. These new results and analyses have been included in the results section and justified in the discussion.

Reviewer 3 Report

Comments and Suggestions for Authors

The concept of using the urinary phosphate/urea (P/U) ratio to identify inorganic phosphate intake is clinically relevant and timely. However, several important limitations should be addressed before the manuscript can be considered for publication:

  1. Sample size and statistical power: Each intervention arm included only six participants, with no reported power analysis, which severely limits the robustness of the findings.

  2. Duration of intervention: The intervention lasted only 3 days. This short exposure period is insufficient to draw conclusions about medium- or long-term effects, particularly in CKD patients.

  3. Absence of biochemical follow-up in healthy volunteers: No blood parameters were measured in healthy participants, which reduces the physiological interpretability of the results.

  4. Reliance on self-reported dietary intake: The method is prone to recall bias and under-/over-reporting. These limitations should be explicitly discussed, and any validation efforts described.

  5. Generalisability: The conclusions extend to broader populations (including pediatric patients and other CKD stages) without direct supporting data. These extrapolations should be toned down.

  6. Presentation of results: Tables are overly detailed and sometimes repetitive. Combining related tables and adding clear, well-annotated figures would improve readability.

  7. Language and structure: The manuscript contains repeated statements and could be streamlined for clarity and conciseness.

Comments on the Quality of English Language

The English is generally understandable but requires editing to reduce redundancy and improve clarity.

Author Response

Response to Reviewers

We thank the reviewers and editors for their constructive and valuable comments. Below we provide a detailed response to each point raised, indicating how we have revised the manuscript accordingly.

Reviewer 3

We sincerely thank Reviewer 3 for the evaluation of our work and for the constructive comments provided.

Respect to the comment “Sample size and lack of power analysis” as noted for Reviewer 2, we now state explicitly that this is a pilot, exploratory study without formal power calculation. This is clarified in both Methods and Discussion (page 3, line 102 and page 17, line 476).

About the comment relative to the duration of the intervention (3 days) we agree that the intervention is short. This is now clearly stated as a limitation, and we specify that results apply only to acute changes in dietary phosphate, not to medium- or long-term effects (page 16 and line 505).

Regarding the comment relative to the “Absence of biochemical follow-up in healthy volunteers” we acknowledge this important point. However, as the healthy volunteers had preserved renal function, we did not expect meaningful changes in circulating phosphate, calcium, or FGF23 within such a short intervention period. FGF23, in particular, is known to respond primarily to sustained phosphate retention rather than to acute dietary fluctuations. For this reason, and to minimize unnecessary invasiveness in healthy individuals, we did not obtain blood samples in this group. We recognize this as a limitation and now emphasize it more clearly in the Discussion, noting that future studies in healthy participants should include serum parameters for completeness (page 16 and line 512)

With respect to the comment on “Reliance on self-reported dietary intake”, we fully acknowledge the inherent risk of recall bias and the potential for under- or overestimation of dietary intake. Nevertheless, it is reasonable to assume that any such inaccuracies would be relatively consistent across the study period, rather than confined to specific time points. In addition, to minimize this limitation, the dietary questionnaires were carefully explained by the principal investigator, who subsequently reviewed each survey with the participants to enhance accuracy and reliability.

About the comment “Generalisability: conclusions extend to broader populations without direct data”. We have revised the Discussion to avoid extrapolation to pediatric or other CKD populations. Statements are now restricted to the studied cohorts, with broader implications suggested only as hypotheses for future research.

Regarding to the comment relative to “Presentation of results: tables overly detailed, repetitive; need for figures”. We appreciate this suggestion and recognize the value of figures for improving readability. However, given the relatively small sample size and the exploratory nature of the study, we considered that presenting the full numerical results in tables would provide greater precision and transparency. In several instances, graphical representation would not add clarity but rather reduce the granularity of the data. To address the reviewer’s concern, we have deleted some Figures and remade others avoiding redundance and we have included new Tables improving readability. We believe this approach ensures both accuracy and clarity.

Finally, respect to the comment relative to “Language and structure: repeated statements, lack of conciseness” we have thoroughly revised the text to streamline the narrative, remove redundancies, and improve clarity.

Round 2

Reviewer 3 Report

Comments and Suggestions for Authors

The authors have responded comprehensively and constructively to all major points raised in the initial review.

  1. Scope and Limitations: The crucial decision to explicitly designate the study as a pilot, exploratory investigation fully addresses the concerns regarding the small sample size and short intervention duration.

  2. Generalizability: The most important change is the restriction of the conclusions in the Discussion section, avoiding unwarranted extrapolation to pediatric or other CKD populations. This was a critical request, and it has been satisfactorily implemented.

  3. Methodological Justification: The authors provide reasonable physiological justification for the omission of serum measurements in healthy volunteers within an acute 3-day intervention framework.

  4. Presentation: Efforts to streamline the text and improve figures/tables, while maintaining numerical transparency for the small dataset, are appreciated.

The revised manuscript represents a significant improvement and is now suitable for publication in Nutrients.

Author Response

Response to Reviewer 2

We thank the reviewer2 and editors for their constructive and valuable comments. Below we provide a detailed response to each point raised, indicating how we have revised the manuscript accordingly.

The revision is much improved the study. Before final decision,  please address the following technical points:
- Presentation of skewed data: Several variables with non-normal distribution are presented as mean ± SD throughout the manuscript. These should be reported as median (IQR), and comparisons performed with non-parametric tests (e.g., Mann–Whitney) rather than parametric ones. Please clarify whether normality was assessed.

We thank the reviewer for this valuable observation. The normality of continuous variables was assessed using the Shapiro–Wilk test. However, given the small sample size (n = 6) and the paired design of the study we used non-parametric tests. As emphasized by Altman (1991, pp. 213–215, Practical Statistics for Medical Research. Chapman and Hall, London), in very small samples, non-parametric methods are generally preferable, even when there is no evidence against normality, because the robustness of parametric assumptions cannot be guaranteed. In line with this recommendation, we employed the Wilcoxon signed-rank test, a well-established non-parametric alternative for paired data. This choice is also consistent with the guidance of Siegel and Castellan (1988, Nonparametric Statistics for the Behavioral Sciences, 2nd ed., McGraw-Hill), who highlight the suitability of rank-based procedures in small-sample settings where distributional assumptions are uncertain. Therefore, we considered a non-parametric approach more appropriate in our setting.

Following Reviewer 2’s suggestion, our results are now reported as median (IQR) in accordance with the recommendation. In addition, the following text has been included in the Statistical section:

Normality was tested for each variable. Nevertheless, considering the small number of subjects per group, we relied on the recommendations proposed by Siegel (1988, Nonparametric Statistics for the Behavioral Sciences, 2nd ed., McGraw-Hill) and Altman ((1991, pp. 213–215, Practical Statistics for Medical Research. Chapman and Hall, London) to ensure a more appropriate statistical interpretation

- Confidence interval formatting: In Table 7, the 95% CIs are currently expressed in a non-standard way (e.g., “±”). To avoid confusion, please present them in the standard format, e.g., estimate (lower CI – upper CI).

In the same way, the 95% confidence intervals in Table 7 have now been reformatted to the standard notation, i.e., estimate (lower CI – upper CI), instead of the previous “±” format.